

# Human-mediated dispersal of *Geniotrigona thoracica* (Apidae: Meliponini) colonies promotes high genetic diversity and reduces population structuring in managed populations

Orawan Duangphakdee[1], Ekgachai Jeratthitikul[2], Pisit Poolprasert[3], Rujira Pongkitsittiporn[3], Chama Inson[3,4] and Atsalek Rattanawannee[3,4]

[1] Native Honeybee and Pollinator Research Center, Ratchaburi Campus, King Mongkut's University of Technology Thonburi, Thung Khru, Bangkok, Thailand
[2] Animal Systematic and Molecular Ecology Laboratory, Department of Biology, Faculty of Science, Mahidol University, Rachadhavi, Bangkok, Thailand
[3] Department of Entomology, Faculty of Agriculture, Kasetsart University, Chatuchak, Bangkok, Thailand
[4] Research and Lifelong Learning Center for Urban and Environmental Entomology, Kasetsart University Institute for Advanced Studies, Kasetsart University, Chatuchak, Bangkok, Thailand

## ABSTRACT

The stingless bee *Geniotrigona thoracica* is a key managed pollinator in Southeast Asia, valued for its honey, propolis, and colony trade. In Thailand, frequent human-mediated movement of colonies raises concerns about its effects on genetic diversity and population structure. We analysed variation in mitochondrial (*COI* and *16S rRNA*) and nuclear (five microsatellite loci) markers from 70 colonies sampled across 17 meliponaries in seven southern provinces. Microsatellite data revealed high genetic diversity and low nuclear differentiation ($K = 1$; $F_{st} = 0.0024–0.1219$; all $P > 0.05$), with extensive gene flow ($N_m = 3.60–207.83$) among provinces. In contrast, mitochondrial markers indicated moderate-to-high differentiation ($F_{st} = 0.619$), consistent with mito-nuclear discordance arising from sex-biased. Managed colonies exhibited elevated heterozygosity and allelic richness, likely reflecting admixture from colony exchange, while unique haplotypes in certain provinces suggest introductions from external sources. Significant inbreeding was detected only in Yala, possibly linked to habitat loss and reduced effective population size. Our findings indicate that current meliponicultural practices maintain high genetic diversity in *G. thoracica* despite mitochondrial structuring, but increasing colony movement between genetically distinct populations may risk erosion of local adaptations, underscoring the need for genetic screening prior to translocation.

Corresponding author
Atsalek Rattanawannee, fagralr@ku.ac.th

# INTRODUCTION

Stingless bees, members of the tribe Meliponini, represent a diverse clade of eusocial Hymenoptera widely distributed across tropical and subtropical regions (*Quezada-Euán, 2018*; *Wongsa et al., 2024*). With approximately 600 described species, they exhibit considerable variation in morphological characteristics, colony structure, and foraging ecology (*Hrncir & Maia-Silva, 2013*; *Quezada-Euán, 2018*; *Rattanawannee & Duangphakdee, 2019*). As dominant pollinators in many tropical ecosystems, stingless bees contribute significantly to the reproductive success of both native flora and cultivated crops (*Heard, 1999*; *Wongsa, Duangphakdee & Rattanawannee, 2023*). Their ecological effectiveness is underpinned by traits such as floral constancy, perennial colony maintenance, reduced defensive behavior due to non-functional stingers, and efficient worker recruitment, all of which enhance their value as pollinators in natural and agroecosystem (*Bartelli, Santos & Nogueira-Ferreira, 2014*; *Wongsa, Duangphakdee & Rattanawannee, 2023*).

During the past two decades, the practice of stingless beekeeping, or meliponiculture, gained increasing traction in Thailand among both commercial and small-scale beekeepers, reflecting its dual role in promoting ecological sustainability and generating supplementary income (*Rattanawannee & Duangphakdee, 2019*). To date, at least 33 stingless bee species across 10 genera have been reported in Thailand, with *Geniotrigona thoracica* emerging as one of the most successfully managed species for commercial purposes (*Rattanawannee & Duangphakdee, 2019*; *Wongsa, Duangphakdee & Rattanawannee, 2023*). This species is well managed to standard wooden hive boxes and is increasingly utilized for pollination services in open-field cultivation of economically important crops. Beyond its pollination role, *G. thoracica* is also valued for its high-yield and high-value production of honey and propolis, as well as for the commercial trade of whole colonies (*Rattanawannee & Duangphakdee, 2019*).

In domestic markets, the honey produced by *G. thoracica* is typically sold at prices ranging from 1,000 to 1,200 Thai Baht (approximately 30–36 USD) per kilogram, substantially higher than that of honey derived from *Apis mellifera*, and at least twice the price of honey from native species such as *A. cerana*, *A. dorsata*, and *A. florea* (*Rattanawannee & Duangphakdee, 2019*). Notably, a premium product known as "emerald honey," which is produced by *G. thoracica* from nectar predominantly collected from *Melaleuca cajuputi* in the southern provinces of Pattani and Narathiwat, can command prices as high as 6,000 Thai Baht (approximately 180 USD) per kilogram (A. Rattanawannee, pers. comm., 2024). Furthermore, fully provisioned *G. thoracica* colonies maintained in wooden hives are commercially valued between 6,000 and 8,000 Thai Baht (approximately 180–242 USD) per colony (A. Rattanawannee, pers. comm., 2024). With increasing consumer demand for high-quality stingless bee products, meliponiculture, particularly involving *G. thoracica*, holds substantial promise as a supplementary livelihood strategy for rural communities throughout southern Thailand.

The commercial exchange and relocation of stingless bee colonies facilitate their movement beyond native geographic boundaries (*Chapman et al., 2018*; *Jaffé et al., 2016b*).

This anthropogenic activity has been associated with ecological concerns, particularly the risk of introducing non-native species into novel environments, which may negatively impact indigenous bee communities and disrupt local biodiversity and ecological functions (*Beekman et al., 2008*; *Byatt et al., 2015*; *Inoue & Yokoyama, 2010*; *Kondo et al., 2009*; *Soland-Reckeweg et al., 2009*). Moreover, translocating colonies across regions increases the likelihood of disseminating parasites and pathogens that can threaten both wild and managed bee populations (*Byatt et al., 2015*; *Chapman et al., 2018*; *Lozier & Zayed, 2017*; *Meixner, Kryger & Costa, 2015*; *Oldroyd & Nanork, 2009*). Understanding the genetic composition and population structure of stingless bees is therefore critical for developing sustainable management strategies (*Koffler et al., 2017*; *Lozier & Zayed, 2017*). Such genetic insights can inform domestication efforts and assist beekeepers in minimizing the risks of inbreeding and genetic erosion, which are common challenges in meliponiculture (*Chapman et al., 2018*).

The movement of bee colonies beyond their native hybrid zones and natural geographic boundaries can lead to genetic consequences such as hybridization and mating interference (*Byatt et al., 2015*). Hybridization involves the genetic exchange between previously reproductively isolated populations (*Byatt et al., 2015*), potentially resulting in the erosion of unique genetic lineages and the homogenization of distinct ecotypes (*Frankham, Ballou & Briscoe, 2010*). This process may ultimately lead to the loss of locally adapted genotypes through genomic swamping (*Frankham, Ballou & Briscoe, 2010*). Mating interference, on the other hand, arises from interspecific reproductive interactions that negatively impact reproductive success (*Byatt et al., 2015*; *De La Rúa et al., 2009*), such as reduced fertility (*Byatt et al., 2015*; *Chapman et al., 2018*; *Koeniger & Koeniger, 2000*; *Remnant et al., 2014*) or unsuccessful mating attempts, thereby diminishing the overall fitness of native bee populations (*Groening & Hochkirch, 2008*).

Previous research has shown that the extent of inbreeding and genetic differentiation among wild and managed stingless bee populations varies considerably across species and geographic regions (*Chapman et al., 2018*; *Landaverde-González et al., 2017*; *Rattanawannee et al., 2020*; *Santiago et al., 2016*). This variability underscores the importance of species and context-specific assessments (*Chapman et al., 2018*), as findings from one system may not be applicable to another. Importantly, empirical evidence suggests that human-mediated management practices may exert a stronger influence on population genetic structure than natural factors such as dispersal ability, habitat loss, elevation gradients, or climatic conditions (*Greenleaf et al., 2007*; *Kükrer, Kence & Kence, 2021*). These observations highlight the need for careful consideration of beekeeping interventions to avoid compromising genetic integrity and local adaptation in stingless bee populations (*Jaffé et al., 2016b*).

The genetic structure of a population is shaped by the dynamic interplay of evolutionary processes, including natural selection, genetic drift, gene flow, and mutation, all of which influence the distribution and frequency of genetic variation within and among populations (*Bluher, Miller & Sheehan, 2020*; *Hartl & Clark, 1997*). In the context of managed pollinators, these forces can be further modified by anthropogenic factors such as selective breeding, artificial colony propagation, and human-mediated dispersal. In this

study, we examined the genetic architecture of *G. thoracica* (Apidae: Meliponini), a stingless bee species of economic and ecological importance that is extensively cultivated for honey production across various regions of Thailand. Our objectives were to determine whether managed populations of *G. thoracica* exhibit significant genetic differentiation from one another, and to evaluate whether they display elevated levels of inbreeding. By analyzing mitochondrial and microsatellite markers, this research provides critical insights into the extent to which meliponicultural practices influence the genetic diversity and structure of *G. thoracica*. These findings have direct implications for the sustainable management and conservation of stingless bees, particularly in the context of colony trade, domestication, and the maintenance of genetically healthy populations for long-term apicultural success.

## MATERIALS AND METHODS

### Sampling and DNA extraction

Seventy adult worker bees were sampled from 17 meliponaries situated in the southern region of Thailand (Table S1; Fig. 1). Each of the bees represented one individual per colony. Collections were performed at the entrance tubes of each nest to ensure colony-specific sampling. All specimens were immediately preserved in absolute ethanol and stored at $-20°$ C prior to laboratory analysis. The geographic coordinates of each sampling site were recorded using a GPS handheld device (Garmin eTrex 20X Handheld GPS). The thorax was used for genomic DNA extraction using a DNeasy® Blood & Tissue kit (Qiagen, Hilden, Germany) following to the instructions of manufacturer.

### Ethics statement

This study did not require any special permits, as it involved no endangered or protected species. Only a limited number of specimens were collected, and all procedures adhered to ethical standards in accordance with established research protocols. Animal handling and experimental methods complied with the ethical guidelines approved by the Animal Experiment Committee of Kasetsart University, Thailand (Approval No. ACKU68-AGR-005).

### Microsatellite analysis

We genotyped a single worker from each colony using a panel of five microsatellite loci, including TC3.302 and TC4.287 originally developed for *Tetragonula carbonaria* (*Green, Franck & Oldroyd, 2001*), A43 and A113 derived from *Apis mellifera* (*Estoup et al., 1995*), and B124 isolated from *Bombus terrestris* (*Estoup et al., 1993*). Polymerase chain reaction (PCR) amplifications were performed according to the published protocols specific to each marker. Amplified fragments were submitted to Macrogen Inc. (Seoul, South Korea) for fragment analysis. The resulting electropherograms were manually inspected and allele sizes were determined using Peak Scanner Software v1.0 (Thermo Fisher Scientific, Waltham, MA, USA), ensuring consistency and accuracy in allele scoring for genetic analyses.

Genetic diversity parameters, including the number of alleles ($N$), the effective number of alleles ($N_e$), and both observed ($H_o$) and expected heterozygosity ($H_e$), were calculated for each population and locus using Option 5 of the GENEPOP software package (*Rousset,*

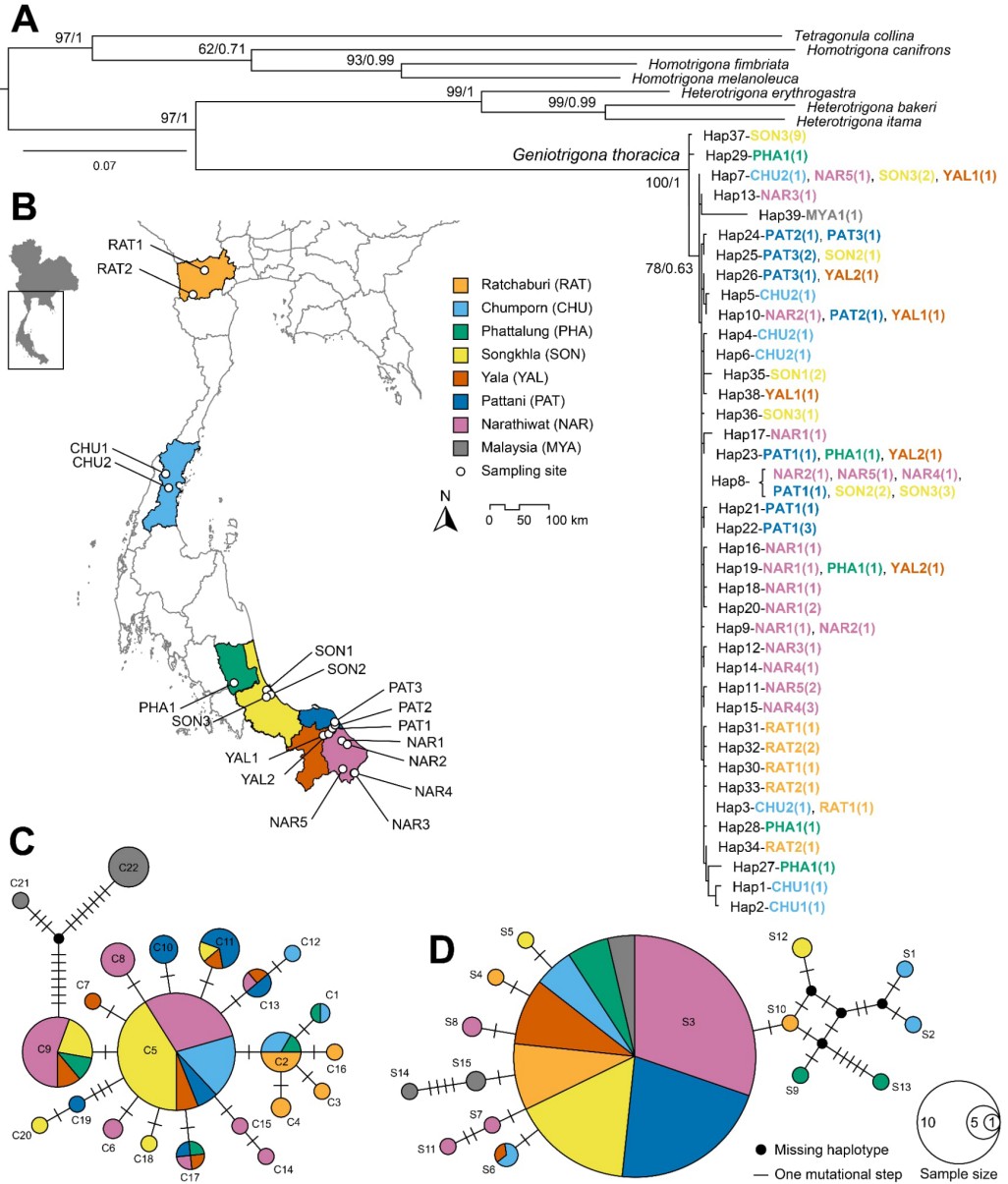

**Figure 1** **Phylogenetic tree and haplotype network analyses.** (A) Maximum Likelihood (ML) phylogenetic tree of *Geniotrigona thoracica* and related species inferred from a 1,149 bp concatenated alignment of mitochondrial *COI* and *16S rRNA* gene sequences. Numerical values at nodes indicate bipartition posterior probabilities (bpp) from Bayesian inference (BI) analysis and bootstrap support values (BS) from Maximum Likelihood analysis, and presented as BI/ML. The scale bar denotes branch length. (B) Map of southern Thailand showing sampling localities of *G. thoracica*, with locality abbreviations corresponding to those listed in Table S1. Median-joining network for *COI* (C) and *16S rRNA* gene sequences (D). Each circle represents a unique haplotype, with the size of the circle proportional to the number of individuals sharing that haplotype. Colors indicate the geographic origin of the samples. The lines connecting the haplotypes represent mutational steps, with each small hash mark signifying a single mutation. The map was created in QGIS v3.24.3 using river and lake layers derived from the HydroSHEDS dataset (https://www.hydrosheds.org) and raster imagery sourced from NASA EARTHDATA (https://www.earthdata.nasa.gov).

*2008*). To evaluate deviations from Hardy–Weinberg equilibrium (HWE), we performed exact tests for HWE and assessed genotypic linkage disequilibrium among loci across populations within GENEPOP.

To investigate the genetic structure of the populations, we employed a Bayesian clustering approach using STRUCTURE version 2.3.3 (*Pritchard, Stephens & Donnelly, 2000*). Analyses were conducted under an admixture model with correlated allele frequencies. Each run comprised a burn-in period of 100,000 steps followed by 1,000,000 Markov Chain Monte Carlo (MCMC) iterations. We tested values of K (the number of genetic clusters) ranging from 1 to 10, with ten independent replicates for each K to ensure consistency. The optimal number of clusters was determined using the ΔK method proposed by *Evanno, Regnaut & Goudet (2005)* as implemented in Structure Harvester (*Earl & von Holdt, 2012*).

To infer gene flow among populations, the effective number of migrants per generation ($Nm$) was derived from the fixation index ($F_{st}$) according to the relationship $Nm = (1 - F_{st})/(2F_{st})$, as described by *Wright (1951)* and further elaborated by *Slatkin (1985)* and *Szalanski et al. (2016)*.

## Mitochondrial DNA analysis

Two mitochondrial gene fragments—cytochrome c oxidase subunit I (*COI*) and large ribosomal subunit rRNA (*16S rRNA*)—were amplified and sequenced by Macrogen Inc. (Seoul, South Korea). Amplification of the *COI* gene employed primers LoboF1 and LoboR1 (*Lobo et al., 2013*), while *16S rRNA* was targeted using primers 16sar-L-myt and 16Sbr-H-myt (*Lydeard, Mulvey & Davis, 1996*). Forward and reverse sequence reads were assembled and manually edited using MEGA11 software (*Tamura, Stecher & Kumar, 2021*). All resulting sequences have been deposited in the GenBank under the accession numbers provided in Table S1. Sequence alignments for each gene were performed independently in MAFFT v.7.49 (*Katoh & Standley, 2013*) by using the L-INS-i algorithm for all genes. Following alignment and trimming, the final sequence lengths were 649 bp for *COI* and 499 bp for *16S rRNA*.

Nucleotide base compositions for the partial *COI* and *16S rRNA* gene sequences were analyzed using MEGA11 (*Tamura, Stecher & Kumar, 2021*). Genetic diversity parameters, including the number of polymorphic sites ($S$), average number of nucleotide differences ($k$), number of haplotypes ($No$), haplotype diversity ($hd$), and average pairwise nucleotide differences ($Pi$), were subsequently calculated using DNAsp version 5.0 (*Librado & Rozas, 2009*).

To infer the historical demographic patterns of *G. thoracica* populations in Thailand, we performed neutrality tests including *Tajima, (1989)*'s *D* and *Fu, (1997)*'s *Fs* using ARLEQUIN v3.5 (*Excoffier & Lischer, 2010*). Tajima's *D* was used to detect departures from neutrality, where positive values may indicate population structure or contraction, and negative values suggest population expansion. Fu's *Fs* was applied to assess the excess of rare alleles, with large negative values interpreted as evidence of the recent population growth. Additionally, Ramos-Onsins and *Ramos-Onsins & Rozas (2002)*'s $R_2$ statistic was calculated in DNAsp v5.0 (*Librado & Rozas, 2009*). The statistical significance of all tests was evaluated using 1,000 coalescent simulations.

To evaluate the impact of stingless beekeeping practices, particularly colony translocation, on the genetic structure of *G. thoracica*, an analysis of molecular variance (AMOVA) (*Excoffier, Smouse & Quattro, 1992*) was conducted using the full mitochondrial dataset in ARLEQUIN version 3.5.2.2 (*Excoffier & Lischer, 2010*). Pairwise $F_{st}$ values were calculated to estimate genetic distances between populations and incorporated into the AMOVA, with statistical significance assessed through 1,000 permutations at a threshold of $\alpha = 0.05$. In addition, $F$-statistics were employed to quantify the extent of genetic differentiation, with significance likewise evaluated using 1,000 random permutations.

Phylogenetic trees were reconstructed using maximum-likelihood (ML) and Bayesian inference (BI) analyses using the dataset of 70 workers of *G. thoracica* collected from southern Thailand and one sample from Malaysia as ingroups, along with other seven stingless bee species as outgroups. Details of the taxon sampling used in the phylogenetic analysis are provided in Table S1. The concatenated alignment was used for unique haplotype identification as implement in DNAsp v5.0. Then, the concatenated alignment of the unique haplotype was divided into four partitions (three partitions for each of three *COI* codons and one partition for *16S rRNA* gene). The best-fit substitution model for each partition was determined using Partition Finder2 v.2.3.4 (*Lanfear et al., 2016*) under the corrected Akaike Information Criterion (AICc). The best-fit model was identified as GTR+I for the first and second codon partitions of *COI*, HKY+G for the third codon partition of *COI*, and GTR+I+G for *16S rRNA*. These models were applied to each gene for subsequent phylogenetic analysis.

All phylogenetic reconstructions were conducted online using the CIPRES Science Gateway platform (*Miller, Pfeiffer & Schwartz, 2010*). The ML analysis was implemented in IQ-TREE version 2.2.2.7 (*Minh et al., 2020*), incorporating 10,000 ultrafast bootstrap replicates (UFBoot) to evaluate the robustness of the inferred topology (*Hoang et al., 2018*). The BI analysis was performed with MrBayes version 3.2.7 (*Ronquist et al., 2012*), utilizing four Markov Chain Monte Carlo (MCMC) chains run for 10,000,000 generations, with sampling occurring every 1,000 generations. All estimated parameters demonstrated effective sample sizes (ESS) exceeding 200. The resulting phylogenetic trees from both BI and ML analyses were visualized and edited using FigTree v.1.4.4 (https://tree.bio.ed.ac.uk/software/figtree/). Clades were regarded as strongly supported when exhibiting ultrafast bootstrap values $\geq$ 95% and Bayesian posterior probabilities $\geq$ 0.95 (*Hoang et al., 2018*; *San Mauro & Agorreta, 2010*).

A haplotype network was constructed using the median-joining algorithm (*Bandelt, Forster & Röhl, 1999*) as implemented in PopART version 1.7 (*Leigh & Bryant, 2015*) to visualize genealogical relationships among mitochondrial DNA haplotypes. This method combines features of minimum-spanning trees and parsimony-based algorithms to generate the shortest and most parsimonious connections between haplotypes (*Bandelt, Forster & Röhl, 1999*). The network illustrates mutational steps between haplotypes and enables the identification of ancestral and derived lineages, thereby facilitating the interpretation of population structure and historical demography. In addition of our 70, six COI sequences and four *16S rRNA* of *G. throracica* collected from Malaysia published elsewhere (*Cameron, Hines & Williams, 2007*; *Jaapar et al., 2025*; *Kek et al., 2017*; *Kwong et*

*al., 2017*; *Rasmussen & Cameron, 2010*) were also included in haplotype network analysis (see list in Table S1).

## RESULTS

### Microsatellite diversity

Table 1 summarizes the genetic diversity parameters, including the total number of alleles ($N_o$), effective number of alleles ($N_e$), allele frequencies, and both observed ($H_o$) and expected ($H_e$) heterozygosity across all loci and populations. Analysis using the Bayesian clustering method in STRUCTURE revealed no distinct genetic structuring among stingless bee subpopulations collected from different provinces. The optimal number of genetic clusters was inferred to be $K = 1$ under the admixture model, as supported by the highest posterior probability [$\text{Ln}(P) = -1,614.68$, $\text{Var Ln}(P) = 9.68$]. Tests for Hardy–Weinberg equilibrium (HWE) indicated significant deviations ($P < 0.05$) in six of the 35 population–locus pairs analyzed. Notably, individuals from Narathiwat province showed a significant heterozygote excess at locus A43. Linkage disequilibrium analysis identified 12 significant pairwise associations among the 105 possible population–locus combinations. However, no consistent patterns of linkage disequilibrium were found among microsatellite loci in the total sample set ($P$-values ranging from 0.0820 to 0.9776), suggesting loci independence across the population.

The local inbreeding coefficient ($F_{is}$) exhibited positive values across all examined populations, with the exception of those from Chumphon ($F_{is} = -0.038$) and Phatthalung ($F_{is} = -0.091$). Notably, a statistically significant excess of homozygosity was observed exclusively in the Yala subpopulation (Table 2).

Pairwise multilocus $F_{st}$ values indicated low genetic differentiation among populations, ranging from 0.0024 between Phatthalung and Pattani to 0.1219 between Ratchaburi and Pattani, with no comparisons showing statistically significant divergence ($P > 0.05$). The highest estimates of genetic differentiation were observed in comparisons involving Ratchaburi, specifically with Pattani (0.1219), Narathiwat (0.1196), and Yala (0.1167) (Table 3). These generally low $F_{st}$ values are likely attributable to extensive gene flow, as evidenced by the estimated number of migrants per generation ($N_m$), which varied from 3.60 to 207.83 across population pairs (Table 3). These results imply that over three reproductive queens are exchanged between populations each generation, indicating substantial levels of interpopulation genetic connectivity.

### Mitochondrial DNA diversity

After trimming the PCR primers, high quality mitochondrial sequences were recovered for both the cytochrome c oxidase subunit I (*COI*; 650 bp) and 16S ribosomal RNA (*16S rRNA*; 489 bp) gene regions. Analysis of nucleotide composition revealed a strong AT-bias characteristic of insect mitochondrial genomes, with A+T contents of 74.9% for *COI* and 77.9% for *16S rRNA*, respectively. Multiple sequence alignments and pairwise comparisons of *COI* sequences identified 14 parsimony-informative sites, consisting predominantly of transitions ($n = 12$; 85.71%) and fewer transversions ($n = 2$; 14.29%). For the *16S rRNA* gene, 10 parsimony-informative sites were observed, comprising four transitions and six
**Table 1** Number of alleles detected ($N_o$), number of effective alleles ($N_e$), observed ($H_o$) and expected heterozygosity ($H_e$) at five microsatellite loci in *Geniotrigona thoracica* populations of Thailand. The number of colonies analyzed from each province is shown in the brackets.

| Locus | Ratchaburi ($n = 7$) | Chumporn ($n = 7$) | Songkhla ($n = 12$) | Phattalung ($n = 5$) | Yala ($n = 6$) | Pattani ($n = 13$) | Narathiwat ($n = 23$) | All Populations ($n = 73$) |
|---|---|---|---|---|---|---|---|---|
| TC3.302 | | | | | | | | |
| $N_o$ | 2 | 4 | 5 | 3 | 6 | 3 | 5 | 8 |
| $N_e$ | 1.153 | 1.849 | 2.286 | 1.852 | 3.999 | 1.977 | 2.713 | 2.308 |
| $H_o$ | 0.143 | 0.429 | 0.667 | 0.6 | 0.667 | 0.308 | 0.478 | 0.466 |
| $H_e$ | 0.133 | 0.459 | 0.562 | 0.46 | 0.749 | 0.494 | 0.631 | 0.567 |
| TC4.287 | | | | | | | | |
| $N_o$ | 3 | 3 | 5 | 2 | 5 | 4 | 6 | 10 |
| $N_e$ | 1.343 | 2.178 | 3.064 | 1.923 | 3.79 | 2.38 | 2.821 | 2.628 |
| $H_o$ | 0.286 | 0.714 | 1 | 0.58 | 0.667 | 0.615 | 0.652 | 0.685 |
| $H_e$ | 0.255 | 0.541 | 0.674 | 0.48 | 0.736 | 0.58 | 0.646 | 0.619 |
| A43 | | | | | | | | |
| $N_o$ | 5 | 4 | 4 | 4 | 3 | 3 | 4 | 6 |
| $N_e$ | 3.062 | 2.8 | 3.097 | 2.941 | 2.88 | 2.086 | 2.829 | 3.543 |
| $H_o$ | 0.571 | 0.429 | 0.583 | 0.6 | 0.67 | 0.615 | 0.478 | 0.507 |
| $H_e$ | 0.673 | 0.643 | 0.677 | 0.66 | 0.652 | 0.521 | 0.646 | 0.718 |
| A113 | | | | | | | | |
| $N_o$ | 3 | 2 | 3 | 2 | 2 | 3 | 3 | 5 |
| $N_e$ | 2.279 | 1.849 | 1.767 | 1.471 | 1.946 | 1.476 | 1.715 | 1.775 |
| $H_o$ | 0.429 | 0.714 | 0.5 | 0.4 | 0.483 | 0.231 | 0.565 | 0.507 |
| $H_e$ | 0.561 | 0.459 | 0.434 | 0.32 | 0.486 | 0.322 | 0.417 | 0.437 |
| B124 | | | | | | | | |
| $N_o$ | 2 | 2 | 3 | 3 | 2 | 3 | 4 | 4 |
| $N_e$ | 1.96 | 1.96 | 2.072 | 2.381 | 1.8 | 2.299 | 2.807 | 2.35 |
| $H_o$ | 0.286 | 0.486 | 0.334 | 0.58 | 0 | 0.615 | 0.609 | 0.566 |
| $H_e$ | 0.489 | 0.489 | 0.517 | 0.48 | 0.444 | 0.565 | 0.644 | 0.574 |

**Table 2** Multilocus microsatellite variation in Thailand's commercial *Geniotrigona thoracica* populations.

| Province | $n$ | $N_o$ | $N_e$ | $H_o$ | $H_e$ | $F_{it}$ | $F_{is}$ |
|---|---|---|---|---|---|---|---|
| Ratchaburi | 7 | $3.00 \pm 1.225$ | $1.959 \pm 0.766$ | $0.343 \pm 0.163$ | $0.422 \pm 0.223$ | 0.122 | 0.162 |
| Chumporn | 7 | $3.00 \pm 1.00$ | $2.127 \pm 0.399$ | $0.554 \pm 0.147$ | $0.518 \pm 0.077$ | 0.074 | −0.038 |
| Songkhla | 12 | $4.00 \pm 1.00$ | $2.46 \pm 0.598$ | $0.617 \pm 0.247$ | $0.573 \pm 0.104$ | 0.085 | 0.033 |
| Phattalung | 5 | $2.80 \pm 0.84$ | $2.114 \pm 0.564$ | $0.552 \pm 0.086$ | $0.480 \pm 0.121$ | −0.182 | −0.091 |
| Yala | 6 | $3.60 \pm 1.82$ | $2.883 \pm 1.015$ | $0.497 \pm 0.289$ | $0.613 \pm 0.141$ | 0.258[*] | 0.224[*] |
| Pattani | 13 | $3.20 \pm 0.45$ | $2.044 \pm 0.356$ | $0.477 \pm 0.191$ | $0.496 \pm 0.103$ | 0.091 | 0.079 |
| Narathiwat | 23 | $4.40 \pm 1.14$ | $2.577 \pm 0.484$ | $0.556 \pm 0.078$ | $0.597 \pm 0.101$ | 0.113 | 0.089 |
| Mean±SD | $10.43 \pm 6.32$ | $3.428 \pm 0.594$ | $2.309 \pm 0.338$ | $0.514 \pm 0.088$ | $0.529 \pm 0.069$ | $0.132 \pm 0.066$ | $0.102 \pm 0.068$ |

**Notes.**

$n$, number of colonies.

The mean observed ($N_o$) and effective ($N_e$) number of alleles, observed ($H_o$) and expected heterozygosity ($H_e$) with standard error (SD), fixation index between individuals and total data set ($F_{it}$), and fixation index between individuals and the local population ($F_{is}$).

*$P < 0.05$.

**Table 3 Pairwise genetic differentiation and estimated gene flow per generation values.** Pairwise genetic differentiation ($F_{st}$) and estimated gene flow per generation ($N_m$) among *Geniotrigona thoracica* apiaries across different provinces in Thailand based on five microsatellite markers.

| | Ratchaburi | Chumporn | Songkhla | Phattalung | Yala | Pattani | Narathiwat |
|---|---|---|---|---|---|---|---|
| Ratchaburi | – | 7.82 | 5.56 | 6.72 | 3.78 | 3.6 | 3.68 |
| Chumporn | 0.0601 | – | 22.12 | 8.86 | 14.7 | 42.6 | 85.7 |
| Songkhla | 0.0825 | 0.0221 | – | 23.89 | 17.81 | 138.39 | 25.69 |
| Phattalung | 0.0692 | 0.0534 | 0.0205 | – | 21.72 | 207.83 | 12.59 |
| Yala | 0.1167 | 0.0329 | 0.0273 | 0.0225 | – | 19.34 | 33.51 |
| Pattani | 0.1219 | 0.0116 | 0.0036 | 0.0024 | 0.0252 | – | 10.39 |
| Narathiwat | 0.1196 | 0.0058 | 0.0191 | 0.0382 | 0.0147 | 0.0459 | – |

transversions. Haplotype analyses revealed high levels of mitochondrial diversity, with 22 unique *COI* haplotypes and 16 distinct *16S rRNA* haplotypes identified. Estimates of genetic diversity indicated high haplotype diversity for COI ($hd = 0.947 \pm 0.012$) and moderate diversity for *16S rRNA* ($hd = 0.405 \pm 0.075$), whereas nucleotide diversity remained low across both loci (COI: $Pi = 0.0034 \pm 0.0003$; 16S rRNA: $Pi = 0.0022 \pm 0.0006$) (Table 4).

The mitochondrial *COI* and *16S rRNA* gene fragments were concatenated into a single alignment comprising 1,139 base pairs. A total of 38 distinct haplotypes were identified across the dataset (Table 4). Of these, 16 haplotypes were shared by at least two individuals, while the remaining 22 haplotypes were singletons, each detected in only one individual. The most frequently observed haplotype (H8) was present in nine specimens, which were sampled from Narathiwat, Pattani, and Songkhla provinces. No significant association was detected between nucleotide diversity ($Pi$) and sample size (Pearson's $r = 0.212$, $P = 0.154$), validating the use of $Pi$ for comparative analyses across populations. Summary statistics for mitochondrial genetic diversity are provided in Table 4.

To evaluate the neutrality of the *G. thoracica* population, summary statistics including Tajima's $D$, Fu's $Fs$, and Ramos-Onsins and Rozas' $R_2$ were calculated, with results summarized in Table 4. When analyzing all specimens collectively, both Tajima's $D$ and Fu's $Fs$ values exhibited a negative but statistically non-significant values ($P > 0.05$) across all mitochondrial genes, indicating an absence of excess rare alleles within the population. Furthermore, the $R_2$ values obtained from the Ramos-Onsins and Rozas test were consistently small and positive across all gene datasets, which is generally consistent with a scenario of recent population expansion in *G. thoracica*.

Upon dividing the samples into seven distinct provincial populations, the majority of Tajima's $D$ and Fu's $Fs$ statistics were negative, while the Ramos-Onsins and Rozas' $R_2$ values were positive. Nevertheless, none of these results reached statistical significance (Table 4). Overall, the findings suggest a lack of clear evidence for recent population expansion among most *G. thoracica* populations in Thailand.

AMOVA results based on mitochondrial gene sequences revealed that the majority of genetic variation occurred among populations within provinces. For the *COI* gene, 30.95% of the variation was partitioned among populations within provinces ($F_{st} = 0.5912$, $P < 0.01$), while 22.21% was attributed to variation within populations. Similarly, for the *16S rRNA* gene, 32.81% of the variation was found among populations within provinces

**Table 4 Summary of molecular diversity indices and population expansion test statistics of mitochondrial cytochrome c oxidase subunit-I (COI) and large ribosomal subunit rRNA gene (16S rRNA) genes.** Number of individuals ($N$), number of haplotypes ($No$), number of polymorphic (segregation) sites ($S$), average number of nucleotide differences ($k$), haplotype diversity ($hd$) and nucleotide diversity ($P_i$) with standard deviation (SD), Tajima's $D$, Fu's $Fs$ and Ramos–Onsins and Rozas' $R_2$.

| Gene | | $N$ | $No$ | $S$ | $k$ | $hd$ ($\pm SD$) | $P_i$ ($\pm SD$) | $D$ | $Fs$ | $R_2$ |
|---|---|---|---|---|---|---|---|---|---|---|
| COI | Province | | | | | | | | | |
| | Ratchaburi | 7 | 4 | 3 | 1.048 | 0.810(0.130) | 0.0016 (0.0004) | −0.654 | −1.390 | 0.170 |
| | Chumporn | 7 | 5 | 5 | 2.041 | 0.905(0.103) | 0.0031(0.0007) | −0.099 | −1.548 | 0.185 |
| | Songkhla | 12 | 6 | 8 | 1.712 | 0.818(0.096) | 0.0026(0.0009) | −1.412 | −1.748 | 0.147 |
| | Phattalung | 5 | 5 | 8 | 3.400 | 1.000(0.126) | 0.0059(0.0013) | −0.807 | −2.004 | 0.174 |
| | Yala | 6 | 6 | 6 | 2.200 | 1.000(0.096) | 0.0034(0.0004) | −0.932 | −1.087 | 0.076 |
| | Pattani | 12 | 8 | 5 | 1.924 | 0.924(0.057) | 0.0029(0.0003) | 0.598 | −1.167 | 0.184 |
| | Narathiwat | 21 | 12 | 9 | 1.933 | 0.933(0.031) | 0.0030(0.0003) | −0.771 | −1.669 | 0.097 |
| | *All samples* | *70* | *22* | *22* | *2.186* | *0.947(0.012)* | *0.0034(0.0003)* | *−1.674* | *−1.159* | *0.047* |
| 16s rRNA | | | | | | | | | | |
| | Ratchaburi | 7 | 3 | 2 | 0.571 | 0.524 (0.209) | 0.0012(0.0005) | −1.237 | −0.922 | 0.2259 |
| | Chumporn | 7 | 4 | 8 | 3.429 | 0.810(0.130) | 0.0070(0.0022) | 0.263 | 0.928 | 0.1873 |
| | Songkhla | 12 | 3 | 4 | 1.076 | 0.439(0.158) | 0.0022(0.0009) | −0.661 | −0.836 | 0.1330 |
| | Phattalung | 5 | 3 | 8 | 3.600 | 0.700(0.218) | 0.0074(0.0028) | −0.440 | −1.674 | 0.2537 |
| | Yala | 6 | 2 | 1 | 0.333 | 0.333(0.215) | 0.0007(0.0004) | −0.993 | −0.950 | 0.3727 |
| | Pattani | 12 | 1 | 0 | 0 | 0 | 0 | / | / | / |
| | Narathiwat | 21 | 4 | 4 | 0.552 | 0.348(0.128) | 0.0011 (0.0005) | −1.434 | −1.296 | 0.1157 |
| | *All samples* | *70* | *16* | *18* | *1.102* | *0.405(0.075)* | *0.0022(0.0006)* | *−2.108* | *−1.853* | *0.0433* |
| Concatenated genes | *Province* | | | | | | | | | |
| | Ratchaburi | 7 | 6 | 5 | 1.619 | 0.952(0.096) | 0.0014(0.0002) | −1.024 | −0.969 | 0.1098 |
| | Chumporn | 7 | 7 | 13 | 5.429 | 1.000(0.076) | 0.0048(0.0011) | 0.126 | 0.304 | 0.1681 |
| | Songkhla | 12 | 6 | 12 | 2.788 | 0.818(0.096) | 0.0024(0.0007) | −1.254 | −0.479 | 0.1135 |
| | Phattalung | 5 | 5 | 16 | 7.000 | 1.000(0.126) | 0.0062(0.0014) | −0.649 | −0.832 | 0.1348 |
| | Yala | 6 | 6 | 7 | 2.533 | 1.000(0.096) | 0.0022(0.0003) | −1.011 | −0.995 | 0.0909 |
| | Pattani | 12 | 8 | 5 | 1.924 | 0.924(0.057) | 0.0017(0.0002) | 0.598 | 0.563 | 0.1842 |
| | Narathiwat | 21 | 14 | 13 | 2.486 | 0.957(0.026) | 0.0022(0.0003) | −1.118 | −0.863 | 0.0912 |
| | *All samples* | *70* | *38* | *40* | *3.288* | *0.970(0.010)* | *0.0029(0.0003)* | *−2.002* | *−1.715* | *0.0395* |

($F_{st} = 0.6491$, $P < 0.01$), with 21.13% occurring within populations (Table 5). When both mitochondrial genes were concatenated, 36.75% of the total genetic variation was observed among populations within provinces ($F_{st} = 0.6186$, $p < 0.01$), whereas variation among provinces and within populations accounted for 27.41% and 24.13%, respectively (Table 5).

Phylogenetic analyses were performed using 39 unique haplotype datasets of *G. thoracica* from Thailand (38 haplotypes) and Malaysia (one haplotype) as ingroups, and other seven bee species as outgroups. Tree topologies derived from both Maximum Likelihood (ML) and Bayesian Inference (BI) approaches were broadly congruent, differing only in the arrangement of terminal clades. Given the similarity, only the ML topology is presented (Fig. 1A). The resulting phylogeny strongly supports the monophyly of *G. thoracica* (bpp = 1.0; BS = 100%), which forms a well-supported sister lineage to *Heterotrigona* species

**Table 5** Analysis of molecular variance (AMOVA) was conducted on *Geniotrigona thoracica* populations using mitochondrial cytochrome c oxidase subunit I (COI) and large ribosomal subunit rRNA gene (16S rRNA) sequences, with populations grouped according to seven distinct geographical provinces in Thailand.

| Gene | Source of variation | df | Sum of squares | Variance components | Percentage of variation | Statistics |
|---|---|---|---|---|---|---|
| **COI** | | | | | | |
| | Among provinces | 6 | 271.787 | 3.9813 | 24.64 | $F_{ct} = 0.2134$ |
| | Among populations within province | 11 | 243.659 | 5.3448 | 30.95 | $F_{sc} = 0.3920^{*}$ |
| | Within population | 60 | 184.135 | 3.5185 | 22.21 | $F_{st} = 0.5912^{**}$ |
| ***16S rRNA*** | | | | | | |
| | Among provinces | 6 | 39.649 | 0.6146 | 31.24 | $F_{ct} = 0.3241$ |
| | Among populations within province | 11 | 28.729 | 0.6179 | 32.81 | $F_{sc} = 0.5818^{**}$ |
| | Within population | 60 | 27.326 | 0.4405 | 21.13 | $F_{st} = 0.6491^{**}$ |
| **Concatenated dataset** | | | | | | |
| | Among provinces | 6 | 489.536 | 5.6717 | 27.41 | $F_{ct} = 0.3314$ |
| | Among populations within province | 11 | 454.426 | 7.9671 | 36.75 | $F_{sc} = 0.4972^{**}$ |
| | Within population | 60 | 368.482 | 4.8685 | 24.13 | $F_{st} = 0.6186^{**}$ |

*indicate significant difference at $p < 0.05$.
**indicate significant difference at $p < 0.01$.

(*H. itama*, *H. bakeri*, and *H. erythrogastra*), with maximal posterior probability (bpp = 1.0) and high, though slightly lower, bootstrap support (BS = 97%). Geographic distribution of all *G. thoracica* samples is shown in Fig. 1B. Within the *G. thoracica* clade, there was no evidence of geographic structuring or genetic divergence. Bees from different geological sampling sites were grouped and mixed within one large clade. Nevertheless, only one subclade was weakly formed with insufficient nodal support (BS = 78%, bpp = 0.63) containing specimens from all sampled provinces and one individual from Malaysia.

Median–joining haplotype networks for *COI* and *16S rRNA* genes (Figs. 1C and 1D) revealed patterns consistent with the corresponding phylogenetic trees, though the 16S rRNA network exhibited lower resolution. The *COI* network comprised 22 haplotypes arranged in a star-like configuration, with two predominant haplotypes (C9, C5) present in all provinces except Ratchaburi. Two haplotypes from Malaysia were slightly separated from the major group. They connected with haplotype C9, differing by 10 mutational steps for haplotype C21 and 14 steps for haplotype C22.

In general, the *16S rRNA* network also showed a star-like configuration. However, most *16S rRNA* haplotypes were confined to single provinces. Only haplotype S3 was found across all provinces, shared by 56 individuals (80% of the total), along with haplotype S6, which was shared between Chumphon and Yala (Fig. 1D).

## DISCUSSION

Previous research has frequently linked artificial selection in managed breeding systems to elevated inbreeding and diminished genetic variation relative to wild progenitors (*Bruford, Bradley & Luikart, 2003*; *Muir et al., 2008*; *Wang et al., 2014*). In contrast, our data show that managed *G. thoracica* populations in Thailand sustain substantial mitochondrial

and nuclear genetic diversity, despite exhibiting pronounced genetic differentiation. This pattern indicates that prevailing stingless bee colony management practices in Thailand exert negligible effects on the overall genetic variability of *G. thoracica.*

Over the last century, apicultural management has significantly shaped the distribution and genetic structure of social bees worldwide, including various species of honey bee, bumble bee, and stingless bee (*Bryant & Krosch, 2016*; *Chahbar et al., 2013*; *Chapman et al., 2018*; *Francisco et al., 2014*; *Jaffé et al., 2016b*; *Jensen et al., 2005*; *Rangel et al., 2016*). The stingless bee *G. thoracica* is a particularly valuable focal species for assessing these impacts, given the rapid expansion of hive trading within Southeast Asia, especially in Thailand and Malaysia (*Rattanawannee & Duangphakdee, 2019*). Anthropogenic hive translocation may provide genetic and adaptive benefits by increasing allelic diversity and facilitating responses to environmental pressures (*Chapman et al., 2018*; *Todesco et al., 2016*; *Wongsa et al., 2024*); however, it can also generate maladaptive hybrids when reproductive barriers exist, and may result in the loss of regionally adapted genotypes (*Byatt et al., 2015*; *Todesco et al., 2016*; *Wongsa et al., 2024*). In southern Thailand, a major center of stingless beekeeping for over two decades, *G. thoracica* populations now exhibit genetic patterns indicative of admixture from multiple geographic origins. While the genetic changes are apparent, their phenotypic implications remain uncertain. Notably, in stingless bees, male attendance at mating aggregations is not always linked to hybridization (*Law et al., 2024*), as illustrated by *Tetragonula carbonaria* males that visit *T. hockingsi* aggregations without exhibiting short-range attraction to the latter's queens (*Paul et al., 2023*).

This study found no evidence that geographical or physical barriers, such as mountain ranges, urban or agricultural landscapes, or forest cover, significantly influence the population structure of Thai *G. thoracica*. Similar patterns, gene flow occurs when the absence of dispersal barriers, have been documented in *Trigona nigerrima*, *Trigona corvina*, and *Scaptotrigona mexicana* in Mexico (*Rodríguez et al., 2024*; *Solórzano-Gordillo et al., 2015*), *Tetragonula carbonaria* and *Tetragonula hockingsi* in Australia (*Brito et al., 2014*; *Law et al., 2024*), *Trigona spinipes* in Brazil (*Jaffé et al., 2016a*) and *Heterotrigona itama* in Thailand (*Wongsa et al., 2024*). The observed structuring of *G. thoracica* populations is more likely driven by the inherently low dispersal capacity of virgin queens, together with ecological variation among local habitats.

Analyses of population genetic structure indicated that several *G. thoracica* populations exhibited limited differentiation from geographically distant groups. Such genetic homogeneity is likely maintained through ongoing gene flow, potentially consistent with a stepping-stone dispersal process (*Kimura & Weiss, 1964*). Anthropogenic factors, particularly the deliberate relocation of colonies by beekeepers, appear to further reinforce interpopulation connectivity. This inference is supported by Bayesian phylogenetic analyses of concatenated mitochondrial COI and 16S rRNA sequences, which grouped 68 of the sampled colonies into a single well-supported clade, with only two colonies forming a separate lineage (Fig. 1A). Moreover, the most common haplotypes, C5 (*COI*) and S3 (*16S rRNA*), were shared across all sampling locations (Figs. 1C and 1D), suggesting extensive haplotype mixing among regions. These results suggest that both natural dispersal and anthropogenic colony translocation are important drivers of population structure in this

stingless bee species. The elevated occurrence of unique haplotypes in Yala and Chumphon further suggests that colonies may have been introduced from other regions, artificially enhancing local genetic diversity.

Across most comparisons, genetic diversity metrics did not differ significantly among groups (Tables 2 and 5). Consistent with our expectations, relatively high levels of genetic diversity, as measured by expected heterozygosity, were detected in all provinces of southern Thailand, where *G. thoracica* colonies are predominantly managed. This pattern was further supported by the elevated values of both expected heterozygosity ($H_e$) and allelic richness ($N_e$) observed in all managed apiaries (Table 2). The enhanced diversity in managed colonies is likely the consequence of admixture over time, driven by the exchange of colonies among beekeepers from different localities, which introduces novel alleles into populations (*Carvalho-Zilse et al., 2009*; *Chapman et al., 2018*; *Wongsa et al., 2024*). This inference is reinforced by pairwise per-generation migration rate (Nm) estimates, all of which exceeded three (ranging from 3.60 to 207.83), indicating substantial queen dispersal among populations (Table 3). Specifically, the data suggest that more than three reproductive queens per generation are exchanged between each pair of populations. In line with these findings, low genetic differentiation ($F_{st}$) values were observed among geographic localities, and AMOVA results for mitochondrial markers revealed no clear geographic partitioning of genetic variation in *G. thoracica* (Table 5). This contrasts with the study of *Rattanawannee et al. (2017)*, which identified two distinct genetic groups of the stingless bee *Tetragonilla collina* in Thailand using geometric morphometric and mitochondrial *COI* sequence analyses. They proposed that, for this subterranean-nesting species, present-day ecological factors, such as seasonal flooding, exert a stronger influence on spatial distribution than historical biogeography.

Although comparative studies between wild and managed stingless bee populations remain scarce, previous work on *Tetragonisca angustula* (*Santiago et al., 2016*) and *Heterotrigona itama* (*Wongsa et al., 2024*) reported no detectable differences in genetic diversity between the two management types. In the present study, nearly all $F_{is}$ values across the defined genetic groups were positive yet statistically non-significant (Table 2). Notably, the absence of significant $F_{is}$ values in managed colonies was unexpected, as such conditions could be indicative of elevated relatedness among colonies within an apiary, a pattern that may arise from colony propagation practices by beekeepers, as previously proposed (*Santiago et al., 2016*). In contrast, the Yala population exhibited positive and significant $F_{is}$ values (Table 2), suggestive of inbreeding, potentially attributable to habitat loss and landscape alterations that may reduce effective population sizes, increase genetic relatedness, and diminish genetic diversity (*Lozier & Zayed, 2017*). To improve understanding of genetic diversity and inbreeding dynamics in *G. thoracica*, broader sampling efforts are required, both in terms of the number of colonies and the range of localities represented. In certain localities, only a single colony was sampled, limiting the precision of diversity estimates. Expanding sample sizes would enable more robust statistical inferences and facilitate the assessment of whether habitat degradation is exerting a negative influence on the genetic diversity of *G. thoracica*.

Bayesian clustering of nuclear genotypes in STRUCTURE supported a single genetic cluster ($K = 1$), whereas AMOVA of concatenated mitochondrial sequences revealed moderate-to-high population differentiation ($F_{st} = 0.619$). This discrepancy exemplifies mito-nuclear discordance, a pattern frequently associated with sex-biased dispersal. In stingless bees, colony founding by queens typically occurs through short-range budding events, averaging approximately 700 m from the natal nest, while males may disperse up to 20 km prior to mating (*Bueno et al., 2022*). Such asymmetry in dispersal capacity can generate stronger genetic structuring in maternally inherited mitochondrial DNA compared to biparentally inherited nuclear loci (*Law et al., 2024*; *Peters et al., 1999*; *Quezada-Euán, 2018*; *Quezada-Euán et al., 2022*). Comparable trends have been reported in *S. mexicana*, where male-biased dispersal has been invoked to explain genetic admixture within drone aggregations (*Rodríguez et al., 2024*). In stingless bee, drones depart their natal colonies to join "drone congregations" situated near nests with virgin queens (*Quezada-Euán, 2018*), which may contain several hundred individuals originating from multiple and often geographically distant colonies (*Dos Santos et al., 2016*; *Kraus, Weinhold & Moritz, 2008*; *Mueller, Moritz & Kraus, 2012*). Although meliponine drones generally exhibit shorter effective dispersal ranges than their honey bee (*Apis* spp.) counterparts (*Kraus et al., 2005*; *Oldroyd et al., 1998*; *Oldroyd & Wongsiri, 2006*), the low genetic differentiation and minimal pairwise genetic distances observed between Chumphon and Narathiwat suggest that ongoing male-mediated dispersal likely contributes to gene flow between these populations (Table 3; Fig. 1). These results indicate that the dispersal distance of stingless-bee drones may be greater than assumed and should be re-evaluated.

In commercial meliponiculture, artificial colony division is commonly employed to increase colony numbers within an apiary. This practice involves transferring a combination of young and old brood combs, together with honey and pollen pots, from a strong donor colony into a new hive box, thereby establishing a daughter colony (*Quezada-Euán, 2018*; *Santiago et al., 2016*). Such management interventions can alter the distribution of mitochondrial haplotypes within a population, with some haplotypes increasing in frequency while others decline or disappear entirely (*Santiago et al., 2016*). In the present study, this pattern was evident in the Chumporn and Pattani populations, which exhibited pronounced population structuring and a reduced number of haplotypes dominated by a few high-frequency variants (Table 4; Fig. 1). Because mitochondrial haplotypes are maternally inherited, they may be transferred between populations if a colony from one source successfully establishes as a daughter colony within another population (*Chapman et al., 2018*), thereby contributing to mitochondrial structuring (*Francisco et al., 2014*).

In natural populations, elevated mitochondrial structure has often been attributed to the short dispersal range of reproductive swarms, a phenomenon reflecting female queen philopatry. This behavior arises because daughter colonies require immediate access to resources, such as propolis and food, provided by the maternal nest to initiate construction of a new hive (*Inoue et al., 1984*). As a result, the limited dispersal of queens constrains gene flow and reinforces population structure (*Santiago et al., 2016*). In managed settings as in our study, repeated colony division from a restricted pool of source colonies within an apiary can produce a genetic pattern analogous to that generated by queen philopatry (*Santiago*

*et al., 2016*). High levels of mitochondrial structuring have similarly been documented in wild populations of multiple stingless bee species, including *Melipona beecheii* (*Quezada-Euán, 2018*; *Quezada-Euán et al., 2007*), *Partamona helleri* (*Brito & Arias, 2010*), *Plebeia remota* (*Francisco & Arias, 2010*; *Francisco, Santiago & Arias, 2013*), *Tetragonula pagdeni* (*Thummajitsakul, Klinbunga & Sittipraneed, 2011*), *Scaptotrigona hellwegeri* (*Quezada-Euán et al., 2012*), *Partamona mulata* (*Brito et al., 2013*), *Melipona subnitida* (*Bonatti et al., 2014*), and *Tetragonisca angustula* (*Francisco et al., 2017*). Additionally, the Australasian clade of stingless bees was recently found to possess an unusually complex mitochondrial genome structure characterized by gene duplications, which appear to contribute to an elevated substitution rate. This phenomenon was first reported in two Australian species (*Françoso et al., 2023*) and was later identified as a shared feature, to varying degrees, across the entire clade (*Li et al., 2024*). The implications of this atypical mitochondrial architecture for population genetic analyses remain poorly understood; however, an accelerated rate of mitochondrial evolution could potentially intensify patterns of population structuring observed in mtDNA.

## CONCLUSION

In conclusion, our results indicate that most *G. thoracica* populations in Thailand exhibit substantial genetic differentiation. While current levels of colony trade and translocation appear not to have disrupted population structure, an escalation of such practices among genetically distinct populations could pose adverse genetic consequences. Preserving the integrity of local gene pools thus requires minimizing genetic admixture. We therefore recommend conducting targeted genetic assessments prior to the introduction of new ecotypes, and ensuring that colony transfers are restricted to populations with demonstrable genetic similarity.

## ACKNOWLEDGEMENTS

We sincerely thank Mr. Preecha Rod-Im, whose involvement was limited to assisting in the planning and conducting of the field sampling survey and providing occasional advice on population-genetic analysis software.

### Funding

This work was supported by the Kasetsart University Research and Development Institute (KURDI) (grant number FF(KU) 51.68), the Department of Entomology, Faculty of Agriculture and the Research and Lifelong Learning Center for Urban and Environmental Entomology, Kasetsart University, Bangkok, Thailand provided laboratory support, and King Mongkut's University of Technology Thonburi (KMUTT), and National Science, Research and Innovation Fund (NSRF) Fiscal year 2024 (grant number FRB670016/0164). The funders had no role in study design, data collection and analysis, decision to publish, or preparation of the manuscript.

## Grant Disclosures

The following grant information was disclosed by the authors:

Kasetsart University Research and Development Institute (KURDI): FF(KU) 51.68.

Department of Entomology, Faculty of Agriculture and the Research and Lifelong Learning Center for Urban and Environmental Entomology, Kasetsart University.

King Mongkut's University of Technology Thonburi (KMUTT).

National Science, Research and Innovation Fund (NSRF) Fiscal year 2024: FRB670016/0164.

## Competing Interests

The authors declare there are no competing interests.

## Author Contributions

- Orawan Duangphakdee conceived and designed the experiments, performed the experiments, authored or reviewed drafts of the article, and approved the final draft.
- Ekgachai Jeratthitikul conceived and designed the experiments, performed the experiments, analyzed the data, prepared figures and/or tables, authored or reviewed drafts of the article, and approved the final draft.
- Pisit Poolprasert performed the experiments, analyzed the data, prepared figures and/or tables, authored or reviewed drafts of the article, and approved the final draft.
- Rujira Pongkitsittiporn conceived and designed the experiments, performed the experiments, authored or reviewed drafts of the article, and approved the final draft.
- Chama Inson performed the experiments, authored or reviewed drafts of the article, and approved the final draft.
- Atsalek Rattanawannee conceived and designed the experiments, performed the experiments, analyzed the data, prepared figures and/or tables, authored or reviewed drafts of the article, and approved the final draft.

## Field Study Permissions

The following information was supplied relating to field study approvals (i.e., approving body and any reference numbers):

Animal handling and experimental methods complied with the ethical guidelines approved by the Animal Experiment Committee of Kasetsart University, Thailand (Approval No. ACKU68-AGR-005).

## DNA Deposition

The following information was supplied regarding the deposition of DNA sequences:

The group 16s rRNA and COI sequences described here are available at GenBank: PV628640 to PV628608 and PV628570 to PV628538, respectively.

## Data Availability

The raw data are available in the Supplementary Files.

## Supplemental Information

Supplemental information for this article can be found online at http://dx.doi.org/10.7717/peerj.20460#supplemental-information.

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
