# Peer review of "Human-mediated dispersal of Geniotrigona thoracica (Apidae: Meliponini) colonies promotes high genetic diversity and reduces population structuring in managed populations"

_PeerJ, doi:10.7717/peerj.20460_

## Round 0.1 · original submission · Minor Revisions

· Academic Editor

Minor Revisions

Dear Dr. Rattanawannee,

Thank you very much for your submission to PeerJ.

As the Academic Editor handling your article, “Human-mediated dispersal of Geniotrigona thoracica (Apidae: Meliponini) colonies promotes high genetic diversity and reduces population structuring in managed populations”, I have carefully reviewed the work and the reviewers’ comments. In my opinion, the manuscript is of interest and merit, but it requires a few Minor Revisions before it can be considered for acceptance.

Best regards,
Armando Sunny

·

Basic reporting

Dear authors,
let me start with the basics, I think you are reporting on an interesting and important aspect. The presentation is clear and unambiguous. The literature is well reflected, although I would be a little bit more cautious to assume that the stingless bee biology of species from Americas is transferable 1:1 to Asian stingless bees. I am aware that much more is done in the Americas, but I suggest that it because clear to the reader that there is an unbalance when it comes to our knowledge about the biology of stingless bees from the Americas and Asia. I have comments and questions in the PDF, but one would be What do we know about the mating frequency of these bees?

Experimental design

The experimental design is straight forward and the question is well-defined. As I said, I have a few comments and questions in the PDF, like why did you only sample one worker per colony?

Validity of the findings

The results are clear and the conclusions drawn are supported by the data. I would like to see the microsat data actually as part of the supplementary

Additional comments

I enjoyed reading the paper, I have comments and questions in the PDF which should be easy to address. Perhaps it is possible to reply less on literature based on data from stingless bees from Americas.

·

Basic reporting

Supplementary Table S1 is well-presented, however the raw data for the microsatellite loci is missing (the column 'Microsattelite' in this Table just shows an X?). Adding the microsatellite alleles per locus for each sample into this table would ensure that all raw data is available in the Supplementary Information.

Experimental design

no comment

Validity of the findings

Overall the data presentation and analyses are excellent, with the only missing part being the raw allele data for the microsatellite genotyping. This could be readily added to Supplementary Table S1.

Additional comments

This study provides valuable data on the genetic structure of a Thai stingless bee and how it may be impacted by human hive translocations, which is an important consideration as interest in stingless bee-keeping continues to grow in many parts of the world.

Below are a few minor comments for the authors' consideration:

L150: This species is now called Tetragonula carbonaria (formerly Trigona carbonaria)

L265-268: Here you report Nm and equate it to the exchange of queens between populations. I suggest introducing this variable in the Methods, along with some explanation of how it approximates migration.

371: Check this sentence for typos – does it mean to say “where gene flow occurs despite apparent dispersal barriers..”?

377: Here it is mentioned that the observed structuring of G. thoracica populations "is more likely driven
by the inherently low dispersal capacity of virgin queens and drones", but later there is acknowledgement that dispersal is highly sex-biased in stingless bees and that males can traverse significant distances. Perhaps consider rephrasing here therefore?

436: Here it is stated "....males may disperse up to 20 km prior to mating (Quezada-Euan 2018)."
I assume the reference here is to the chapter "Reproduction" pp 131–165 of Quezada-Euan's book. Although that chapter includes an excellent paragraph on male mating aggregations, I could find no mention of the 20 km figure in there. Please check the citation, or consider also citing the below study, which uses empirical data and models to estimate 20 km as the maximum dispersal distance for Australian stingless bees:

Bueno et al. 2022. Males are capable of long distance dispersal in a social bee. Frontiers in Ecology and Evolution. https://doi.org/10.3389/fevo.2022.843156

476: You might consider mentioning also that the Austral-Asian clade of stingless bees was recently discovered to have a highly unusual mitogenome structure involving gene duplications, which seems to result in a higher-than-usual substitution rate. This was first reported in two Australian species (Francosco et al 2023) and subsequently found to be a feature of the whole clade to varying degrees (Li et al 2024). You could perhaps highlight that the implications of this for the population genetics of mtDNA in the clade are not yet well-understood, though a higher rate of mt-evolution would likely amplify a pattern of high population structuring in mtDNA. The relevant papers are below:

Francosco et al 2023. Rapid evolution, rearrangements and whole mitogenome duplication in the Australian stingless bees Tetragonula (Hymenoptera: Apidae): A steppingstone towards understanding mitochondrial function and evolution. Int J Biol Macromol 10.1016/j.ijbiomac.2023.124568

Li et al 2024. Comparative analyses of mitogenomes in the social bees with insights into evolution of long inverted repeats in the Meliponini. Zool Res. 10.24272/j.issn.2095-8137.2023.169

---

## Round 0.2 · accepted · Accept

· Academic Editor

Accept

Dear Author’s,

Thank you for submitting your manuscript to PeerJ. After careful review and consideration of the revisions, I am pleased to inform you that your manuscript has been accepted for publication.

The reviewers and I found your study to be a valuable and well-executed contribution to our understanding of stingless bee population genetics and the effects of human-mediated dispersal on genetic diversity. Your work will be of significant interest to researchers in ecology, conservation, and evolutionary biology.

On behalf of PeerJ, I would like to congratulate you and your co-authors on this accomplishment. We look forward to seeing your article published soon.

Warm regards,

Armando Sunny

·

Basic reporting

no comment

Experimental design

no comment

Validity of the findings

no comment

Additional comments

I congratulate the authors on an interesting and valuable study.